

# Effects of xylo-oligosaccharide and flavomycin on the immune function of broiler chickens

Lin Yuan[1], Wanli Li[1], Qianqian Huo[2], Chenhong Du[2], Zhixiang Wang[2], Baodi Yi[1] and Mingfa Wang[1]

[1] Henan Academy of Agricultural Sciences, Henan Key Laboratory of Farm Animal Breeding and Nutritional Regulation, Institute of animal Husbandry and Veterinary Medicine, Zhengzhou, Henan, China
[2] Henan Agricultural University, College of Animal Science and Veterinary Medicine, Zhengzhou, Henan, China

## ABSTRACT

This study investigated the effects of xylo-oligosaccharide (XOS) and flavomycin (FLA) on the performance and immune function of broiler chickens. A total of 150 ArborAcres broilers were randomly divided into three groups and fed for six weeks from one day of age in cascade cages. The diets of each test group were (1) a basal diet, (2) the basal diet supplemented with 2 mg/kg FLA, and (3) the basal diet supplemented with 2 mg/kg XOS. At 21 and 42 days, the growth performance index values and short-chain fatty acid (SCFA) concentrations in the cecum were quantified. Furthermore, immunoglobulin G (IgG) and plasma interleukin 2 (IL-2) as well as mRNA expression of LPS-Induced TNF-alpha Factor (*LITAF*), Toll-like receptor-5 (*TLR5*) and interferon gamma (*IFNγ*) in the jejunum were quantified. The results showed that administration of XOS or FLA to chickens significantly improved the average daily gain. Supplementation with XOS increased acetate and butyrate in the cecum, while FLA supplementation increased propionate in the cecum. An increase in plasma IgG was observed in XOS-fed 21-day-old broilers, but FLA supplementation decreased IgG in the plasma of 42-day-old broilers and increased plasma IL-2. Furthermore, FLA or XOS supplementation downregulated mRNA expression of *IFNγ*, *LITAF* and *TLR5*. The above data suggest that addition of XOS and FLA to the diet could improve the growth performance of broilers and reduce the expression of cytokine genes by stimulating SCFA.

## INTRODUCTION

Antibiotic growth promoters (AGPs) could improve growth performance and disease resistance and have been widely used in poultry diets (*Castanon, 2007*). However, due to growing concern about the spread of antibiotic-resistant genes in human and animal pathogens, the European Union has limited or even prohibited the use of antibiotics in the poultry breeding industry since 2006 (*Maron, Smith & Nachman, 2013*). However, poultry feed without AGPs may increase the incidence of disease and lead to an increase in the use of antibiotics for treatment (*Casewell et al., 2003*). Therefore, it is necessary to find an

Corresponding author
Mingfa Wang,
wangmingfa2008@126.com

effective alternative to AGPs to ensure the health of poultry as well as the safe and efficient production of poultry products.

FLA is absorbed poorly after oral administration and thus highly valued as a feed additive growth promoter in livestock animals with negligible safety concerns with very low residual concentrations in the tissue of food animals (*Pfaller, 2006*). Studies have shown that AGPs mainly act on intestinal microbes to improve their metabolic activity and immune function (*Dibner & Richards, 2005*; *Oakley et al., 2014*), and the complex microbial community in the intestine interacts with the host animal and affects its health. Therefore, it is possible to adjust the intestinal flora through the use of probiotics to affect the intestinal and overall health of an animal (*Fukuda & Ohno, 2014*). Studies have shown that the probiotic agents that have regulatory effects on microbes are primarily *Bifidobacterium spp.* and *Lactobacillus spp.* (*Gibson, 2004*; *Holck et al., 2011*; *Thomassen et al., 2011*; *Vigsnaes et al., 2011*), and they have been proven to have beneficial effects in animals (*Rastall, 2007*). The final product of the fermentation of dietary fiber by bacteria is mainly short chain fatty acids (SCFA), but this process also provides the host with a variety of prebiotics that regulate intestinal inflammation (*Arpaia et al., 2013*; *Smith et al., 2013*).

Previous studies have shown that xylo-oligosaccharide (XOS)-selective fermentation can improve host health by inducing structural changes in the intestinal flora, indicating that XOS meets the definition of a prebiotic (*Gibson et al., 2004*). The study on mice show that XOS feeding decreases systemic inflammation, and this effect is most likely caused by higher SCFA concentrations as a result of an increased bifidobacterial saccharolytic fermentation in the entire gut and not only in the large intestine, in the intestinal epithelium, expression of interleukin 1$\beta$ (*Il1$\beta$*) ($P < 0.01$) and interferon gamma (*IFN$\gamma$*) ($P < 0.05$) was significantly less in blood from XOS-fed mice than from control-fed mice. In vitro treatment of blood with propionate significantly decreased *Il1$\beta$* ($P < 0.01$), *IFN$\gamma$* ($P < 0.05$), and interleukin 18 (*Il18*) ($P < 0.001$) expression, increased production of SCFA in the gut, which are transported across the intestine and into the systemic compartments, results in downregulation of low-grade inflammatory cytokines (*Hansen et al., 2013*). IFN $\gamma$

In conclusion, it is important to study the effect of XOS on the relative expression level of immune genes in the intestines of broiler chickens. Therefore, the current study assessed the effects of antibiotic flavomycin (FLA) and XOS on cecal SCFA concentrations, the plasma levels of immunoglobulin G (IgG) and interleukin 2 (IL-2) and the expression of *IFN$\gamma$*, LPS-Induced TNF-alpha Factor (*LITAF*) and Toll-like receptor-5 (*TLR5*) in the jejunum of broilers to explain possible beneficial effects on gastrointestinal health.

## MATERIALS AND METHODS

### Chickens and diets

One hundred fifty one-day-old broilers (Ross-308) were divided into three treatment groups (five replicates, 10 chickens each) as follows: (1) CTL: a basal corn-soybean meal diet as a control, (2) FLA: diet 1 supplemented with flavomycin (2 mg active ingredient/kg feed), and (3) XOS: diet 1 supplemented with XOS (2 mg/kg feed). The compounds were uniformly mixed in the basal diet. Broilers were reared in battery cages with raised wire

**Table 1  Diet compositions and nutrient levels.**

| Parameter | 1–21 d | 22–42 d |
| --- | --- | --- |
| Ingredients, % | | |
| Corn | 55.63 | 60.17 |
| Soybean meal | 38.00 | 33.00 |
| Limestone | 1.10 | 1.10 |
| Dicalcium phosphate | 1.80 | 1.60 |
| Table salt | 0.30 | 0.30 |
| DL-Methionine | 0.17 | 0.13 |
| Vegetable oil | 2.00 | 2.70 |
| Premix[a] | 1.00 | 1.00 |
| Nutrient levels | | |
| Metabolic energy, kcal/kg | 2,866.80 | 2,945.64 |
| Crude protein | 22.07 | 20.13 |
| Met, % | 0.33 | 0.31 |
| Lys, % | 1.19 | 1.06 |
| Cys, % | 0.36 | 0.33 |
| Calcium, % | 0.91 | 0.85 |
| Non-phytate phosphorus, % | 0.44 | 0.40 |

**Notes.**

[a]Premix provided per kg of diet: vitamin A, 15,000 IU; vitamin D3, 3,000 IU; vitamin E, 20 IU; VK, 3 mg; thiamine, 4 mg; riboflavin, 8 mg; vitamin B5, 40 mg; pyridoxine, 4.5 mg; vitamin B12, 0.02 mg; pantothenic acid, 30 mg; niacin, 35 mg; choline, 1,300 mg; folic acid, 1.2 mg; biotin, 0.18 mg; Cu (copper sulfate), 8 mg; Fe (ferrous sulfate), 80 mg; Zn (zinc sulfate), 75 mg; Mn (manganese sulfate), 100 mg; I (potassium iodide), 0.4 mg; Se (sodium selenite) 0.15 mg.

flooring, received feed and water ad libitum for 42 days, and were provided artificial light for 24 h. The temperature remained at 33–35 °C in the first week and was then gradually decreased to 24 °C after three weeks. The diets were fed in mash form and formulated according to NRC requirements (Table 1), and the animal use protocol followed the Guide for the Care and Use of Laboratory Animals as adopted and promulgated by the United National Institutes of Health and approved by the Institutional Animal Care and Use Committee of the Henan Academy of Agricultural Sciences (Protocol no. 20170327).

## Sample collection

After fasting for 12 h, broilers were weighed at 21 and 42 days, and the average daily feed intake (ADFI), average daily gain (ADG) and feed to gain ratio (FGR) were recorded and calculated on a cage-by-cage basis on days 21 and 42.

At 21 and 42 d of age, one chick was selected from each replicate, 3 ml of blood were taken from the wing vein, placed in the anticoagulant tube, and centrifuged at 1,000× g for 15 min to separate the plasma and then stored at −20 °C until further analysis. The chickens were anesthetized and then killed by jugular bleeding. The cecal chyme and mid-jejunum (5 cm from Meckel's diverticulum) were immediately frozen in liquid nitrogen and then stored at −70 °C.

## Cecal SCFA concentrations

Two grams of cecal chyme were placed in a centrifuge tube followed by the addition of 5 ml of ultra-pure water, vortex shock for 5 min, and $5,000\times$ g centrifugation for 10 min. Next, 1 ml of the supernatant was placed in a plastic ampere tube; 0.2 ml of 25% phosphoric acid solution was added; and the resulting solution was placed in an ice bath for 30 min followed by $10,000\times$ g centrifugation for 10 min. Then, the supernatant was analyzed on a Hewlett Packard Agilent 6,890 gas chromatograph (Agilent, Palo Alto, CA, USA) to determine the concentrations of cecal chyme acetic acid, propionic acid and butyric acid.

## Measurement of plasma immunological parameters

The plasma concentrations of IgG (No. E026) were spectrophotometrically measured using commercial diagnostic kits, and the IL-2 (No. H003) concentrations were quantified using commercially available ELISA test kits (Nanjing Jiancheng Biotechnology Institute, Nanjing, P.R. China).

## Determination of the mRNA expression of immune genes

The relative expression levels of *IFNγ*, *LITAF* and *TLR5* were detected by RT-PCR. We extracted total RNA from approximately 50 mg of the mid-jejunum using TRIzol reagent (Takara, Dalian, China), and the purity of the isolated RNA was evaluated as previously described (*Wang et al., 2012*). Total RNA samples were prepared according to the instructions for the commercial kit (TaKaRa, Shiga, Japan) used to prepare the reverse transcription reaction solution. The total system was 10 μL, and the reverse transcription product (cDNA) was stored at −20 °C. The primers were synthesized by the Sangon Biological Engineering Technology & Service Co., Ltd. (Shanghai, P.R. China), and the jejunum *IFNγ*, *LITAF* and *TLR5* primer pairs were designed based on the *IFNγ*, *LITAF* and *TLR5* coding sequences, respectively. The glyceraldehyde-3-phosphate dehydrogenase (GAPDH) primer pair was designed based on the conserved GAPDH sequence (Table 2), and real-time quantitative PCR was performed on a Bio-Rad CFX96™ Real-Time PCR System using the EvaGreen dye method. The 20-μL reaction system was as follows: SsoFast™ EvaGreen® Supermix×10 μL (Bio-Rad, Hercules, CA, USA), 1.0 μL of the upstream and downstream primers (100 μmol/L), 2.0 μL of the cDNA template, and the addition of sterilized deionized water to 20 μL. The samples were packed in 96-well plates (Bio-Rad, USA), and the reaction conditions were as follows: 95 °C for 10 s to pre-denaturation, 95 °C for 5 s to denaturation, and 60 °C for 34 s to annealing and extension. The quantity of immune gene mRNA in each sample was normalized to GAPDH. The cDNA of the immune genes was quantified using relative standard-curve methods, and the mRNA levels were quantified by the comparison threshold method (*Livak & Schmittgen, 2001*).

## Statistical analysis

The data were analyzed using SAS 8.0 statistical software. The results were expressed as the mean ± standard deviation, and pairwise differences were determined by Duncan's test. Differences were considered statistically significant at $P < 0.05$.

**Table 2 Primer pairs for *IFN$\gamma$*, LITAF, TLR5 and GAPDH genes from broilers.** The glyceraldehyde-3-phosphate dehydrogenase (GAPDH) primer pair was designed based on the conserved GAPDH sequence.

| Primer | Sequence | Fragment length (bp) |
|---|---|---|
| *IFN$\gamma$* | P1 5′-TGAGCCAGATTGTTTCGATG-3′<br>P2 5′-TCCTTTTGAAACTCGGAGGA-3′ | 246 |
| *LITAF* | P3 5′-TGTGTATGTGCAGCAACCCGTAGT-3′<br>P4 5′-GGCATTGCAATTTGGACAGAAGT-3′ | 229 |
| *TLR5* | P5 5′-TGTGGGAGAGAGGTTTATGTTTGG-3′<br>P6 5′-CTGAGAGAGAGGTGAGACAATAGG-3′ | 169 |
| GAPDH | P7 5′-CTACACACGGACACTTCAAG-3′<br>P8 5′-ACAAACATGGGGGCATCAG-3′ | 244 |

**Table 3 Effect of FLA and XOS on performance of broilers.** Each value represents the mean SD of 5 replicates. In the same row, values with no superscript letter or the same superscript letter are not significantly different ($P > 0.05$); those with different superscript letters are significantly different ($P < 0.05$).

| Parameter | CTL | FLA | XOS |
|---|---|---|---|
| 1–21 days | | | |
| ADFI (g/d) | $50.31 \pm 1.23$ | $51.31 \pm 1.66$ | $50.89 \pm 1.35$ |
| ADG (g/d) | $32.95 \pm 0.79$ | $33.81 \pm 0.95$ | $33.44 \pm 1.07$ |
| FCR | $1.53 \pm 0.01$ | $1.52 \pm 0.01$ | $1.52 \pm 0.02$ |
| 1–42 days | | | |
| ADFI (g/d) | $96.46 \pm 1.26^{b}$ | $98.36 \pm 1.59^{b}$ | $101.06 \pm 1.95^{a}$ |
| ADG (g/d) | $50.29 \pm 1.04^{b}$ | $51.83 \pm 1.32^{a}$ | $53.25 \pm 0.88^{a}$ |
| FCR | $1.92 \pm 0.02$ | $1.90 \pm 0.02$ | $1.90 \pm 0.01$ |

**Notes.**
ADFI, average daily feed intake; ADG, average daily gain; FCR, feed conversion ratio.

## RESULTS

### Performance

The growth performance results are presented in Table 3. At 1–21 days, supplementary FLA or XOS had no effect on the average daily feed intake (ADFI), the average daily gain (ADG) or the feed conversion ratio (FCR) ($P > 0.05$). Compared to the control at 1–42 days, ADFI was significantly enhanced with XOS supplementation ($P = 0.001$); ADG was significantly enhanced with FLA ($P = 0.047$) or XOS ($P = 0.001$) supplementation; but supplementing FLA ($P = 0.08$) or XOS ($P = 0.074$) had no effect on FCR compared to the control (Table S1).

### SCFA in the cecum

The cecal SCFA results are presented in Fig. 1. At days 21 and 42, the cecal acetate concentration was significantly higher in the control group than in the FLA group ($P < 0.001$), but was significantly lower than in the XOS group (21 d: $P = 0.003$, 42 da: $P < 0.001$). The cecal propionate concentration was significantly higher in the FLA group than the control group (21 d: $P = 0.001$, 42 d: $P < 0.001$) and the XOS group (21 d:

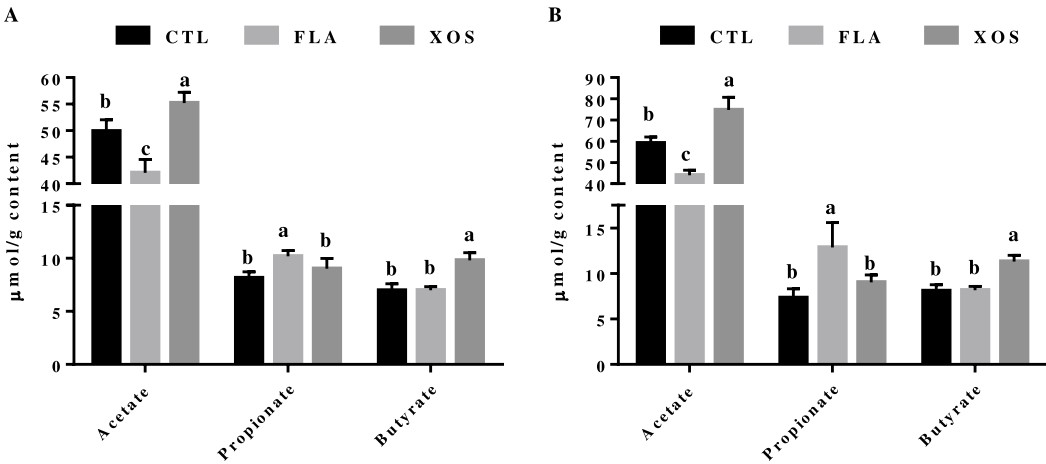

**Figure 1** **Short-chain fatty acid concentrations in the cecum ($\mu$mol/g).** Acetate, propionate and butyrate concentrations ($\mu$mol/g content) collected on 21 (A) and 42 (B) days of age. In the same row, values with no superscript letter or the same superscript letter are not significantly different ($P > 0.05$); those with different superscript letters are significantly different ($P < 0.05$).

**Table 4** **Immune-related factors in the plasma of broiler chickens.** Each value represents the mean SD of five replicates. In the same row, values with no superscript letter or the same superscript letter are not significantly different ($P > 0.05$); those with different superscript letters are significantly different ($P < 0.05$).

| Parameter | CTL | FLA | XOS |
|---|---|---|---|
| IgG (mg/ml) | | | |
| 21 days | $2.03 \pm 0.16^b$ | $2.19 \pm 0.13^b$ | $2.45 \pm 0.21^a$ |
| 42 days | $1.52 \pm 0.11^a$ | $1.31 \pm 0.11^b$ | $1.59 \pm 0.18^a$ |
| IL-2 (ng/L) | | | |
| 21 days | $170.65 \pm 9.32^a$ | $147.84 \pm 16.40^b$ | $171.67 \pm 16.97^a$ |
| 42 days | $215.38 \pm 21.01^a$ | $158.27 \pm 24.01^b$ | $227.29 \pm 24.07^a$ |

$P = 0.025$, 42 d: $P = 0.005$) on days 21 and 42. Cecal butyrate was higher for the XOS group than for the CTL ($P < 0.001$) or FLA ($P < 0.001$) groups at both 21 and 42 d (Fig. S1).

## Immune-related factors in the plasma

The results for immune-related factors in the plasma of broiler chickens are presented in Table 4. At 21 days of age, the IgG in the XOS group was significantly higher than in the control ($P = 0.002$) and FLA ($P < 0.039$) groups, and at 42-day-age, the IgG in the FLA group was significantly lower than in the control ($P = 0.035$) and XOS ($P = 0.008$) groups. The IL-2 in the FLA group was significantly lower ($P < 0.05$) than in the control (21 d: $P = 0.03$, 42 d: $P = 0.002$) and XOS (21 d: $P = 0.024$, 42 d: $P < 0.001$) groups on days 21 and 42 (Table S2).

## Relative mRNA expression of immune genes

The results for relative mRNA expression of immune genes are presented in Fig. 2. Compared to the control group, the FLA and XOS treatments had decreased mRNA levels

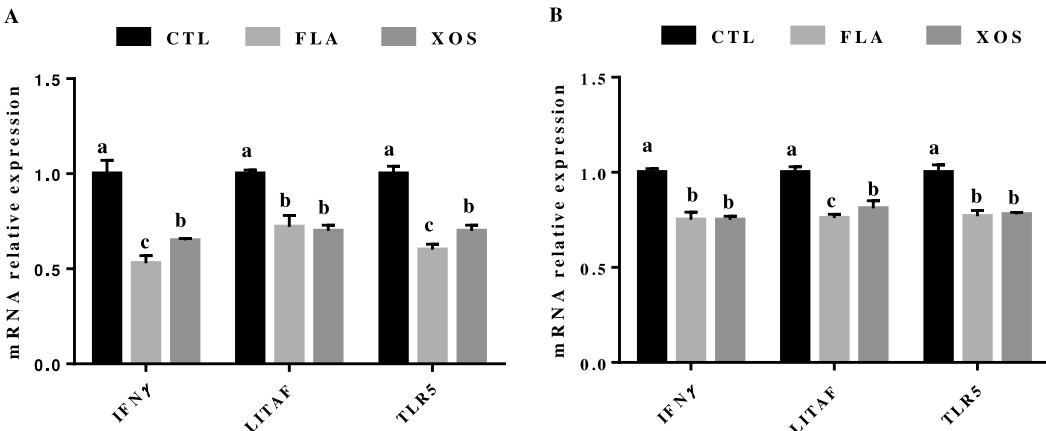

**Figure 2** **The relative mRNA expression of immune genes in the jejunal tissues of broiler chickens.** *IFNγ*, *LITAF* and *TLR5* mRNA relative expressions collected on 21 (A) and 42 (B) days of age. In the same row, values with no superscript letter or the same superscript letter are not significantly different ($P > 0.05$); those with different superscript letters are significantly different ($P < 0.05$).

of jejunum *IFNγ*, *LITAF* and *TLR5* ($P < 0.001$) on days 21 and 42. At 21 day of age, the jejunum *IFNγ* ($P = 0.002$) and *TLR5* ($P = 0.001$) mRNA levels were significantly higher in the FLA group than in the XOS group, and at 42 days of age, the jejunum *LITAF* mRNA level was significantly higher ($P = 0.013$) in the XOS group than in the FLA group (Fig. S2).

## DISCUSSION

As a prebiotic, XOS can be used to selectively alter the structure and activity of the gastrointestinal flora by selective fermentation, thereby improving host health (*Gibson et al., 2004*), and FLA can prevent animal diseases and promote growth and thus is widely used in poultry feed. In this study, broilers with XOS- and FLA-supplemented diets had greater ADG than those of the control group, which agrees with the studies by *De Maesschalck et al. (2015)* and *Zhenping et al. (2013)*. We speculate that may be attributed to the regulation of XOS on intestinal microbes, there is significant difference in the Firmicutes phylum (1.3-fold higher) in the cecum content of the XOS-fed mice than measured in the control-fed mice (*Hansen et al., 2013*), when the proportion of Firmicutes phylum in the intestine increases, the ability of the host to absorb the energy in the food will be strengthened, and the storage of fat in the body will be increased (*Ley et al., 2005*). These results show that adding XOS to the diet can improve the biological function of chickens, suggesting that XOS could be considered an emerging prebiotic, which is defined as a food ingredient that has no nutritional value but can improve the health of the host by regulating its microflora.

Previous studies have shown that XOS is not readily digestible by the gastrointestinal tract or that it has low digestibility overall, ensuring that it can reach the colon (*Imaizumi et al., 1991*; *Nakakuki, 1993*). Although XOS is not degraded by gastrointestinal digestive

enzymes, it can be fermented by intestinal microbes to produce SCFA (*Kabel et al., 2002*). Our results demonstrate that XOS could increase the levels of acetate and butyrate in the cecum of broilers, SCFA may be the key factor in the mechanisms behind the immune-modulatory effect of XOS seen systemically (*Hansen et al., 2013*). The level of plasma IgG in the XOS group of broilers significantly increased compared with the other two groups at 21 days, suggesting that XOS could improve broiler immune function. Furthermore, the plasma IL-2 concentration of the FLA test group was lower than that of the control group; IL-2 is a proinflammatory cytokine, which can promote the growth, proliferation and differentiation of T cells (*Yang et al., 2015*). Interleukins belong to a class of cytokines that function in the immune system by playing an important physiological and pathological role in the inflammatory response process. A cytokine secretion imbalance or cytokine process disorders may lead to a variety of pathological disorders (*Tayal & Kalra, 2008*), indicating that the broilers in the FLA group had less of an inflammatory reaction during growth. Previous research has also found that the addition of virginiamycin in broilers significantly increases the concentration of propionate in the cecum (*Pourabedin, Guan & Zhao, 2015*). In poultry, SCFA enhances intestinal, non-specific immune mechanisms by stimulating the expression of mucin glycoprotein in intestinal epithelial cells to combat pathogens, thus affecting growth performance (*Willemsen et al., 2003*; *Timbermont et al., 2010*). Butyrate is the main source of energy for colon cells and can have anti-inflammatory effects through a number of mechanisms (*Hamer et al., 2008*). Butyrate can inhibit the expression of nuclear factor kappa B (NF-$\kappa$B), which is an important transcription factor that regulates the expression of proinflammatory cytokines (*Inan et al., 2000*). Butyrate can also interfere with the signal transduction of interferon gamma (*IFN$\gamma$*) by inhibiting the activation of signal transducers and the activator of transcription 1 (STAT1) (*Klampfer et al., 2003*). Studies have also shown that butyrate can enhance the expression of peroxisome proliferator-activated receptor gamma (PPAR-$\gamma$), a transcription factor that belongs to the nuclear hormone receptor family that can reduce the expression of inflammatory cytokines and promote immune cell differentiation and this has an anti-inflammatory function (*Martin, 2009*; *Schwab et al., 2007*; *Wächtershäuser, Loitsch & Stein, 2000*).

The intestinal immune system has a complicated relationship with intestinal microorganisms (*Chow et al., 2010*; *Liu et al., 2017*); intestinal commensal flora could effectively inhibit the intestinal inflammatory response and promote immune tolerance through the Toll-like receptor pathway (*Round et al., 2011*). *Pourabedin, Guan & Zhao (2015)* and *De Maesschalck et al. (2015)* determined the effect of XOS and virginiamycin on the microbes in the intestine of broiler chickens by 16S rRNA pyrophosphate sequencing, and the results showed that the addition of XOS and virginiamycin to diets could regulate the relative abundance of intestinal flora. In this study, broilers fed FLA and XOS significantly downregulated the mRNA levels of jejunal *LITAF*, *IFN$\gamma$* and *TLR5* compared to the controls. *LITAF* is a proinflammatory cytokine that is mainly produced by macrophages and monocytes and is involved in normal inflammatory and immune responses. *LITAF* has been shown to increase under many pathological conditions such as in the presence of bacterial toxins, IL-1, IL-2, IFN and other cytokines as well as sepsis, malignancies, heart failure and many others (*Ghareeb et al., 2013*). *TLR5* can bind to bacterial flagellin

and activate proinflammatory cytokines for a proinflammatory response. A previous study showed that host recognition of flagellin can accelerate the aggregation of neutrophils, thereby protecting the host from pathogens (*Ferro et al., 2004*; *Vijaykumar et al., 2007*). The results of this study indicate that the addition of XOS and FLA in the diet can improve the immune function of broilers by reducing the expression of cytokine genes in the jejunum of broilers.

## CONCLUSION

Overall, the present findings showed that the addition of XOS and FLA to the diet can improve the growth performance of broilers. The resulting production of SCFA by supplementing XOS and FLA may thus be the indirect cause of the reduced expression of *LITAF*, *IFNγ* and *TLR5*.

### Funding

The authors received no funding for this work.

### Competing Interests

The authors declare there are no competing interests.

### Author Contributions

- Lin Yuan conceived and designed the experiments, performed the experiments, analyzed the data, contributed reagents/materials/analysis tools, prepared figures and/or tables, authored or reviewed drafts of the paper, approved the final draft.
- Wanli Li conceived and designed the experiments, performed the experiments, analyzed the data, contributed reagents/materials/analysis tools, authored or reviewed drafts of the paper, approved the final draft.
- Qianqian Huo and Chenhong Du performed the experiments, authored or reviewed drafts of the paper, approved the final draft.
- Zhixiang Wang conceived and designed the experiments, analyzed the data, prepared figures and/or tables, authored or reviewed drafts of the paper, approved the final draft.
- Baodi Yi performed the experiments, authored or reviewed drafts of the paper, approved the final draft.
- Mingfa Wang conceived and designed the experiments, contributed reagents/materials/analysis tools, authored or reviewed drafts of the paper, approved the final draft.

### Animal Ethics

The following information was supplied relating to ethical approvals (i.e., approving body and any reference numbers):

The animal use protocol followed the Guide for the Care and Use of Laboratory Animals as adopted and promulgated by the United National Institutes of Health and approved by the Institutional Animal Care and Use Committee of the Henan Academy of Agricultural Sciences (Protocol no. 20170327).

## Data Availability

The raw data is provided in the Supplemental Files.

## Supplemental Information

Supplemental information for this article can be found online at http://dx.doi.org/10.7717/peerj.4435#supplemental-information.

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
