# Peer review of "Effects of xylo-oligosaccharide and flavomycin on the immune function of broiler chickens"

_PeerJ, doi:10.7717/peerj.4435_

## Round 0.1 · original submission · Major Revisions

The reviewers have brought up a number of issues that must be fixed during revision of the paper. Please ensure that everything is addressed prior to re submission.

·

Basic reporting

1. In most journals, animal mRNA names should be italicized e.g. Infγ (Italic, isn't supported on this webpage). I am not sure the formatting of PeerJ.
2. It is controversial to use gene name Tnfα in chickens and some researchers don’t think chickens have one. If you search on NCBI, you will see Litaf with alias of Tnfα. You’d better to write it as “Litaf (Tnfα)” at the first time if you really want to use Tnfα.
3. At line 135, you wrote “mean± standard deviation”, but at Table 3 legend, you wrote “mean± SE”. Please keep consistent across your manuscript.
4. At line 73, please add “antibiotic” before “flavomycin”.

Experimental design

Generally, the experiment design is reasonable. The mRNA data at jejunum are weak because most microbial activities and host immune response are in the low gut (ileum, ceca and large intestine).

Validity of the findings

1. The authors’ findings are relevant to poultry industry. It is consistent that using antibiotic flavomycin (FLA) improved body weight gain and reduced IgG and IL-2 (a sign of lower inflammatory status), but it is interesting that FCR wasn’t improved by FLA.

2. The discussion section is better be more concise and focus on what is closely related to your data in the result section. For example: from lines 176 to 180, the authors don’t have any data related to XOS degradation/digestion and it distracts readers from your main points. From lines 194 to 211, the authors discussed various bacteria, SCFAs, XOS, and FLA. The authors don’t have any data on the bacteria. It would be better to use 1-2 sentences for this paragraph.

3. The authors suggested that increased body weight gain in XOS chickens was resulted from increased SCFAs and reduced immune response, but it wasn’t much convincing. Butyrate doesn’t improve body weight gain in healthy chickens as showed by Zhang et al., Br Poult Sci. 2011 Jun 1;52(3):292-301. The authors showed jejunal immune genes but the main site should be ceca in chickens. The IgG and IL-2 are comparable between CTL and XOS birds at 42 days of age, suggesting similar immune status. The author may read works about XOS and inflammation from Hansen et al. J Nutr. 2013 Apr;143(4):533-40, or other related papers.

Additional comments

Please improve your writing, below are some examples:
At line 37, “not adding AGPs to poultry feed” changes to “poultry feed without AGPs”
At line 62-63, “during which proteins are secreted that regulate immune responses by ..” changes to “where secreted proteins regulate immune response on.. ”
At line 82, “Broilers were reared for 42 days, were free to feed and drink water” changes to “Broilers were received feed and water ad libitum for 42 days”
At line 144, “supplementary” changes to “supplementing”

Reviewer 2 ·

Basic reporting

Abstract
Lines 29-30: “reduced their susceptibility to disease” This is not a proper interpretation for this study in which no disease challenge was used or specific disease diagnosed.

Introduction
The introduction could be improved to more clearly introduce the hypothesis and justify the experimental design. There is a lot of general information included that does not necessarily strengthen the justification of the experimental design. There needs to be more direct emphasis on the test compounds and their potential effects. Why was flavomycin chosen as the compound compare with XOS?
Additional changes to the Introduction below:
Line 44: Change “Therefore, it is possible that adjust…” to “Therefore, it is possible to adjust…:
Line 46: It seems that “prebiotic” should be “probiotic”
Line 48: “proved” should be “proven”
Lines 48-52: The last 2 sentences of this paragraph could probably be combined and/or streamlined to improve flow and clarity.
Line 65: The sentence beginning “When cytokines..” is poorly phrased and can be improved.
Line 76: “Explain possible beneficial effects of antibiotics…” XOS is not an antibiotic, so this should be revised accordingly.

Materials and Methods
What breed cross of broilers was used?
Details on housing/cages of the broilers are needed. This has implications on the outcomes of the experiment as stated below.
How were the compounds added to the basal diet? “On-top” of the basal diet?
Was there any mortality? If so, the amount should be reported. Was the feed conversion ratio corrected for mortality?

Results
Please state the actual P-values throughout the results section.
Line 143: “the highest ADG occurred with supplementary XOS…” The ADG of FLA and XOS were not significantly different, so this statement should be revised.
Lines 151-152: “Supplementary FLA or XOS had no effect on cecal butyrate compared to the control.” This is incorrect – according to Figure 1, cecal butyrate was higher for the XOS group than for the CTL or FLA groups at both 21 and 42 d.
Line 164: “jejunum TNFa mRNA level was significantly higher in the FLA group than in the XOS group” This is incorrect – according to Figure 2, the TNFa was higher for the XOS group than for or FLA group at 42 d.
Tables and Figures
The actual P-values should be listed in the table and figure descriptions, rather than P < or > 0.05. The should be accomplished by having a P-value column to correspond with each measurement.
Table 1
- “ME” should be defined and reported as a whole number.
Table 3
- The table description states that values are means of 6 replications, whereas the number of replicates in the Methods section was identified as 5. Please address.

Discussion
The discussion is somewhat disjointed and does not accurately provide solid interpretations of the results, backed by relevant citations. It should be revised extensively. Some specific changes include:
The inherent fiber content and “fermentability” of the diet used should be addressed? Would same results be expected in a diet with higher fiber ingredients that are commonly used in broiler production?
Line 167-176: Address implication of feed intake response in opening summary paragraph.
Line 196: “reproduction of probiotics” It is likely the authors meant beneficial intestinal bacteria rather than “probiotics” per se
Line 201-202: The effect of XOS on butyrate and acetate production has already been mentioned in Line 182-184. All discussion should regarding XOS and be combined into one subsection of the discussion, and this discussion referenced when necessary in describing other effects.
Line 217: NFkB is a transcription factor, not a cytokine.
Lines 233-238: The discussion of deoxynivalenol is irrelevant to the current experiment and should be eliminated.

Experimental design

Overall, the experimental design is adequate to test the effects of XOS or FLA on the selected outcomes, but several issues should be addressed:

It appears that the chickens were reared in battery cages with raised wire flooring – this should be clearly defined in the Methods section. It is only listed in the abstract as “cascade” cages.

If chickens were in fact reared on raised wire flooring, the significant responses to flavomycin, and to some extent XOS, are surprising. Much of the interpretation is based on potential changes in the intestinal microbiota, but the environmental exposure to bacteria of birds in wire cages is much lower and different than that for birds reared on litter floors under commercial conditions. The authors should provide further info and potential implications of selected housing system in the interpretation of results.

It would have been interesting to see the plasma responses of the same cytokines that were measured by gene expression in the intestine.

Validity of the findings

The difference between the 1-42d ADFI values for FLA (98.36) and XOS (101.06) is only 1.5x the SEM; thus, it seems unlikely that this difference would be statistically different. The same could be said for the 1-42 ADG differences between CTL and FLA (~1.3x the SEM).

There were no differences in FCR, only AFDI and ADG. Indeed, the XOS group ate the most feed and gained the most weight. However, the authors do not address this in their interpretations in the discussion sections. The improved ADG with XOS supplementation is attributed to “biological function” (Line 173), when the data indicate it is merely a function of higher feed intake. The potential implications of XOS on SCFA production and feed intake should be thoroughly addressed in the discussion.

---

## Round 0.2 · accepted · Accept

The authors did a good job addressing the concerns raised.

·

Basic reporting

The authors made relevant changes.

Experimental design

The authors made relevant changes.

Validity of the findings

The authors made relevant changes.